materials science/nanotechnology

ZnO, doping, visible light photocatalyst, electrospinning, annealing

**Author for correspondence:**
Lifeng Cui
e-mail: lcui@dgut.edu.cn

This article has been edited by the Royal Society of Chemistry, including the commissioning, peer review process and editorial aspects up to the point of acceptance.

# Facile fabrication of Mn²⁺-doped ZnO photocatalysts by electrospinning

Yuting Wang[1,2], Xin Hao[4], Zegao Wang[3,5], Mingdong Dong[4] and Lifeng Cui[1]

[1]School of Environment and Civil Engineering, Dongguan University of Technology, Guangdong 523808, People's Republic of China
[2]Department of Engineering, Materials and Polymer Engineering, Nanofiber Technology and Cellular Engineering, Menglin Chen group, and [3]Interdisciplinary Nanoscience Center, Aarhus University, DK-8000 Aarhus C, Denmark
[4]North Laser Research Institute Co. Ltd, 610000, Chengdu, People's Republic of China
[5]College of Materials Science and Engineering, Sichuan University, 610054 Chengdu, People's Republic of China

MD, 0000-0002-2025-2171

In this study, a high-efficiency photocatalyst was synthesized by Mn²⁺-doped ZnO nanofibres (NFs) fabricated by facile electrospinning and a following annealing process, in which Mn²⁺ successes incorporate to ZnO NFs lattice without changing any morphology and crystalline structure of ZnO. The photodegradation properties of ZnO loading with different concentrations of Mn²⁺ (5, 10, 15 and 50 at%) were investigated. The 50% MnO–ZnO composite owns excellent active photocatalytic performance (quantum efficiency up to 7.57%) compared to pure ZnO (0.16%) under visible light and can be considered as an efficient visible light photocatalyst material. We systematically analysed its catalytic mechanism and found that the enhancement belongs to the Mn doping effect and the phase junction between MnO and ZnO. The dominant mechanism of Mn doping leads to the presence of impurity levels in the band gap of ZnO, narrowing the optical band gap of ZnO. In addition, doped Mn²⁺ ions can be used as electron traps that inhibit the recombination process and promote electron–hole pair separation. In summary, this paper provides a convenient method for fabricating highly efficient visible light photocatalysts using controlled annealing.

## 1. Introduction

Due to the low energy consumption, low cost, low toxicity and no secondary pollution, photocatalytic degradation has been considered as a promising strategy to address the current environmental issue [1–4]. Photocatalysts that can effectively reduce

the barrier in photocatalytic reactions have attracted attention [5,6]. $TiO_2$ has been considered as a photocatalyst candidate, but its low efficiency in the visible light range limits its application [7]. Although the visible light response of $TiO_2$-based catalysts can be improved by doping, the efficiency is still low. Therefore, the development and design of high-efficiency new photocatalysis has become a strategy to solve the low efficiency of visible light catalysis [8].

Compared with $TiO_2$, ZnO shows unique characteristics, such as direct and wide band gap in the near ultraviolet spectrum, strong oxidizing ability, large free exciton binding energy [9] and good photocatalytic performance. It has become an efficient and promising candidate in green environmental management systems. However, its wide band gap (3.37 eV) [10] indicates that it can only work in the ultraviolet range, which greatly limits its photocatalytic efficiency. Currently, there have been many studies on enhancing the visible light photocatalytic activity of ZnO by doping or introducing surface vacancies [3,11–16]. It was found that nanofibres show higher catalytic activity compared to nanoparticles or bulk materials because of their high surface to volume ratio and different crystallinity. Lately, various methods such as hydrothermal, electrospinning and chemical precipitation have been used to prepare photocatalysts with high efficiency. However, ZnO nanocatalysts prepared by these methods generally exhibit larger crystal sizes or lower sample yields. In comparison, electrospinning is considered to be a more practical and facile method for preparing nanofibres with high surface area and small crystal size. Therefore, this technology has proven to be a preferred approach in recent years due to its viable and simple characteristics [2,17–21].

Another important feature that promotes photocatalytic performance is to reduce the driving force that traps electrons on the doping centre, thus they will show better electron transferring and electron–hole pair separation. High-spin $d^4$ ions such as $Mn^{3+}$ provide a huge driving force to capture nearby electrons when converted into a $d^5$ configuration [1]. By contrast, ions such as Fe(III), Mn(II) and Cu(I) capture electrons without obtaining exchange energy, making them an ideal candidate for active dopants [22–27]. Kanan & Carter reported that 1 : 1 alloy of MnO and ZnO shows a very low band gap of 2.6 eV and is an ideal visible light catalytic material for both water oxidation and $CO_2$ reduction reactions [1]. In particular, $Mn^{2+}$ ions with a semi-filled electronic configuration exhibit excellent catalytic activity [22]. They can efficiently capture the charge carriers to promote charge carrier removal of the catalyst surface and accelerate the rate of charge transfer between the interfaces. Thus, in this study, $Mn^{2+}$ is used since it can produce more intermediate states and does not function as a recombination centre.

Because of the difficulty in nanostructure formation and synthesizing $Mn^{2+}$ doping, here, we report a high-efficiency visible light photocatalyst of $Mn^{2+}$-doped ZnO nanofibres through a facile electrospinning and controlled annealing process. The MnO–ZnO composite photocatalyst shows an excellent performance (quantum efficiency up to 7.57%) compared to pure ZnO (0.16%) under visible light. Then, we systematically analysed the catalytic mechanism. The results show that the enhancement belongs to the Mn doping effect and the phase junction between MnO and ZnO.

# 2. Materials and methods

## 2.1. Fabrication

ZnO nanofibres (NFs) added $Mn^{2+}$ (0, 5, 10, 15 and 50 at% $Mn^{2+}$) were fabricated by an electrospinning approach. First, a certain amount of manganese acetate and zinc acetate were dissolved in deionized water. Then, polyvinyl pyrrolidone (PVP) powder and ethanol were mixed and stirred to prepare a PVP solution (0.4 g MW = 130 000) at a concentration of 8 wt%. Finally, the two prepared solutions were mixed and stirred at room temperature until they were thoroughly mixed. Subsequently, the electrospinning is operated with a positive voltage of 15 kV and a distance of 15 cm between the tip and the collector at a flow rate of 0.5 ml $h^{-1}$. Finally, the nanofibre sample was taken from the collecting substrate and thermally annealed using a furnace tube. The prepared sample was annealed at a heating rate of 10°C $min^{-1}$ and maintained at 800°C for 1 h in a controlled $NH_3$ atmosphere, which prevented the transfer of the doping element $Mn^{2+}$ to $Mn^{4+}$.

## 2.2 Characterization

The morphologies of $Mn^{2+}$-doped ZnO NFs were tested using scanning electron microscope (SEM, JSM-7001F, JEOL, Tokyo, Japan). The phase of the prepared samples was characterized using a transmission electron microscope (TEM, JEM-2010F, JEOL, Tokyo, Japan) and a diffractometer (XRD, D/max-2500, Rigaku, Tokyo, Japan).

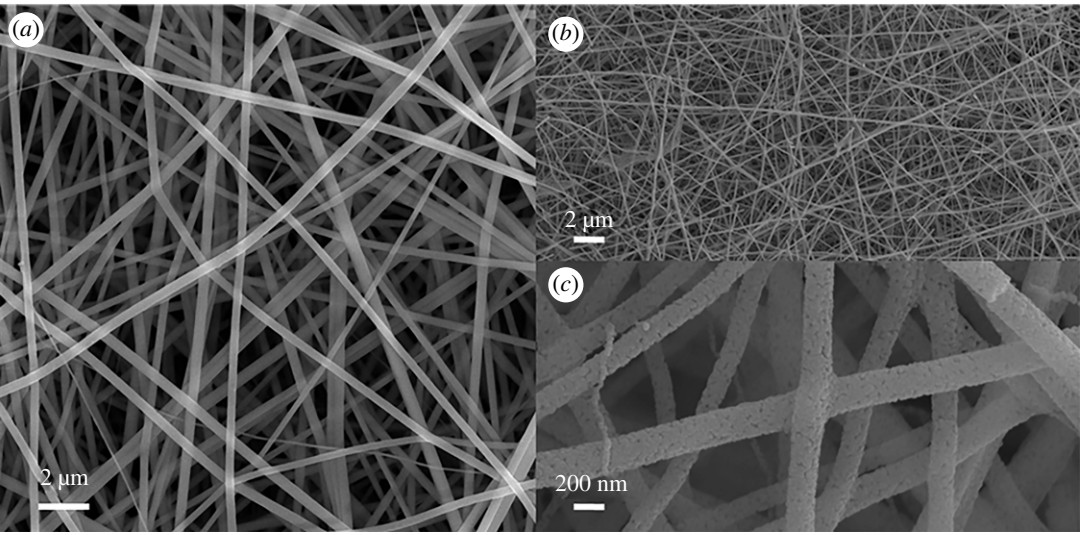

**Figure 1.** SEM images of (*a*) as-spun nanofibres, (*b*) the MnO–ZnO nanofibres annealed at 800°C for 1 h in NH$_3$ and (*c*) enlarged view of the fibres.

## 2.3 Photocatalytic performance

The prepared fibres were dispersed in a Rhodamine B (RhB) aqueous solution ($2.5 \times 10^{-5}$ M, 10 ml) and stirred in the dark for more than 2 h to obtain an adsorption/desorption equilibrium of the sample. The photocatalytic activities of all fibres were evaluated by the degradation of RhB under a Xe lamp with 300 W at a distance of approximately 5 cm and an average visible light intensity of 100 mW ml$^{-1}$, which is assembled with a 400 nm cut-off filter. Upon irradiation, the resulting solution with an interval of 10 min was measured by an ultraviolet–visible spectrophotometer (Shimadzu, UV3600).

# 3. Results and discussion

## 3.1. Characterization

Using electrospinning, uniform fibres with high density were collected on an aluminium substrate. After annealing, the fibres maintained continuous morphology, but the surface turned rough. Figure 1*a* shows a SEM image of as-spun nanofibres with a diameter about 600 nm. All fibres exhibited uniformity and high density and collecting on an aluminium substrate using electrospinning. After annealing for 1 h under an NH$_3$ atmosphere, the fibre maintained its continuous structure having only a rough surface due to crystallization. The diameter of the annealed fibres was decreased to $300 \pm 20$ nm, shown in figure 1*b*,*c*.

The internal structure was further studied by TEM observation. The nanofibres were prepared by uniformly dispersing in ethanol and drop cast on TEM grid. After annealing, the nanofibres showed a pronounced fibrillary structure (figure 2*a*). The inset of figure 2*a* corresponding to the selected area electron diffraction (SAED) pattern shows that the annealed Mn$^{2+}$-doped ZnO nanofibres exhibit high-quality polycrystals with no preferential orientation. Based on the intensity of the wurtzite ring, the presence of (002) (101) (100) and (103) lattice spacing can be obtained. Energy dispersive X-ray spectroscopy (EDS) analysis confirmed that ZnO, O and Mn elements were uniformly present in the nanofibres, and the ratio of M and O elements was approximately 1 : 1 (figure 2*b*). Figure 2*c* shows a high-resolution TEM (HRTEM) image of the fibre and the lattice constants of 0.25 nm and 0.255 nm, corresponding to the ZnO (101) and the MnO (111) planes (marked by a white rectangle in figure 2*a*). In the inset of figure 2*c*, MnO–ZnO nanofibres possess the (001) and (111) lattice plane of ZnO and MnO, which proved the phase junction between MnO and ZnO. Based on the HRTEM and EDS results, hexagonal ZnO and MnO phases are present in the annealed MnO–ZnO samples. XRD patterns of MnO–ZnO nanofibres annealed at 800°C are shown in figure 3*a*, demonstrating their composition and crystal structures. The main reflection peak indicates that the sample is mainly composed of a hexagonal wurtzite ZnO phase. There is no MnO phase in the 0, 5, 10 and 15% Mn$^{2+}$-doping samples, but there was a clear MnO phase peak in the MnO–ZnO composite. The lattice value

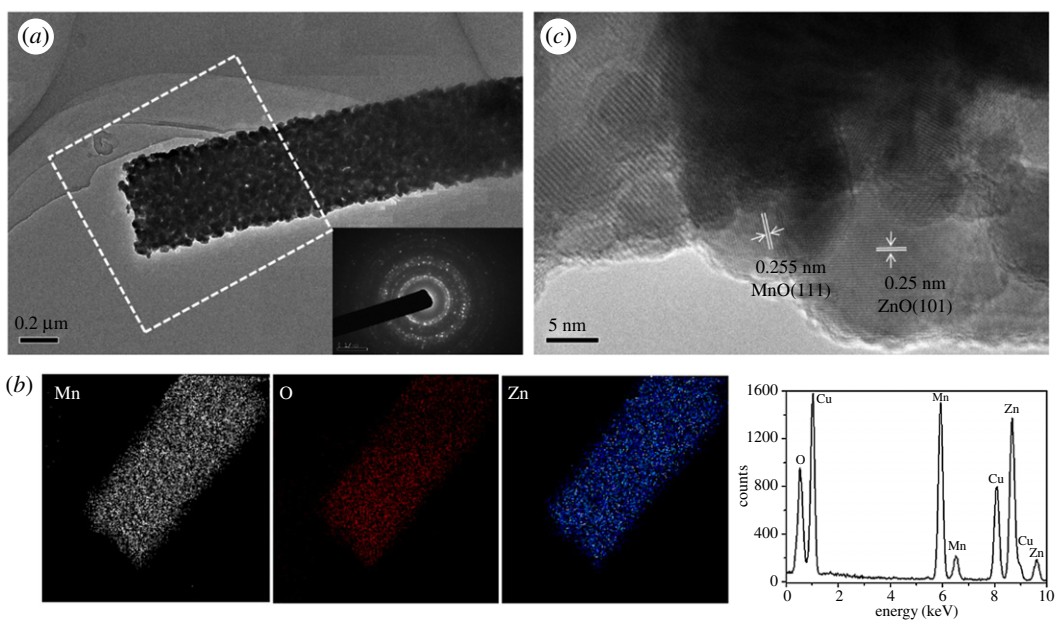

**Figure 2.** (*a*) TEM image, (*b*) EDX pattern and (*c*) HRTEM images of MnO–ZnO nanofibres.

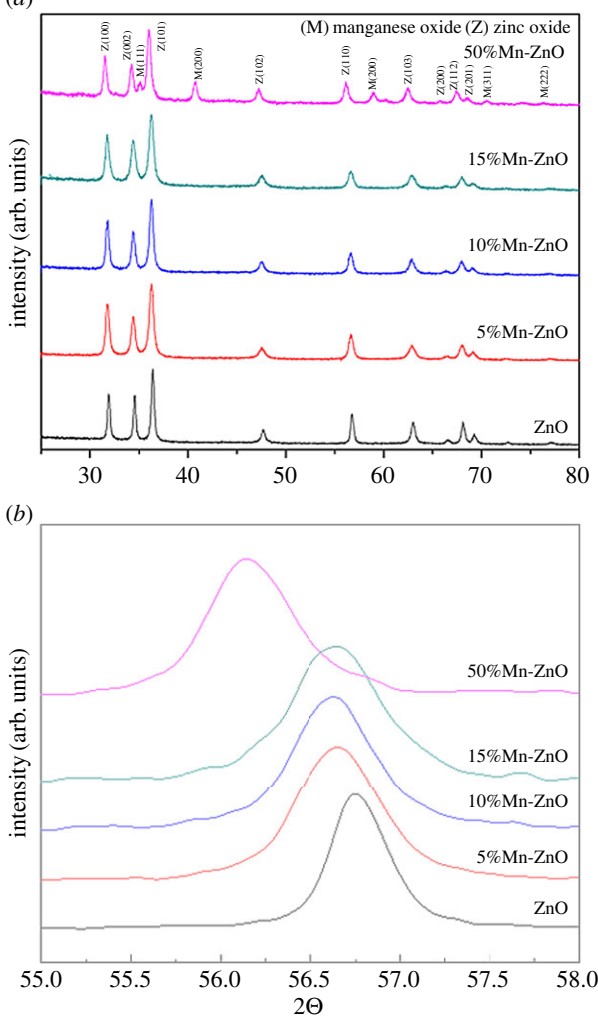

**Figure 3.** (*a*) XRD patterns of ZnO fibres with addition of 5, 10, 15 and 50% Mn²⁺ nanofibres. (*b*) Enlarged XRD patterns in the range from 30° to 40°.

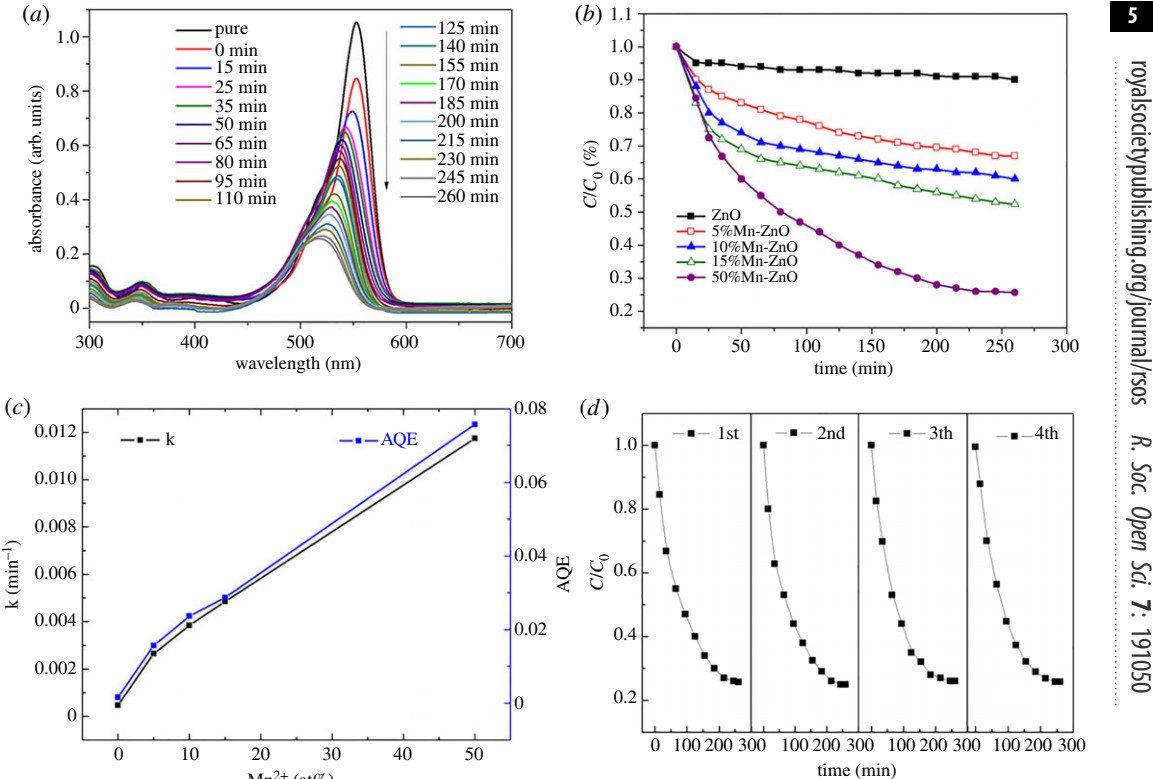

**Figure 4.** (a) Absorbance patterns of MnO–ZnO nanofibres after different time intervals under visible light. (b) The absorption spectra of RhB degraded by ZnO nanofibres doped with different $Mn^{2+}$ concentration. (c) Degradation rate constants and apparent quantum efficiencies (AQE) for $Mn^{2+}$-doped ZnO nanofibres. (d) Cycling performances of the MnO–ZnO specimen.

of ZnO doped with different $Mn^{2+}$ concentration can be calculated by the following Bragg's Law ($2d\sin\theta = n\lambda$) using Jade 6.5 software [28]. With the doping of $Mn^{2+}$, the lattice of ZnO increase from $a = b = 3.242$ Å, $c = 5.194$ Å to $a = b = 3.259$ Å, $c = 5.224$ Å. As known, Mn readily donates its electrons to O, and the preferred substitution position of Mn in the ZnO lattice is the Zn site. From the step scanning XRD patterns in figure 3b, as the Mn concentration increases, the reflection peak shifts to a lower angle in the range of 55.2–58.0°. This phenomenon can be explained by the following. $Mn^{2+}$ ions prefer to occupy the $Zn^{2+}$ positions, resulting in an increase of the lattice parameters and cell volume. Since the ionic $Mn^{2+}$ atom (80 pm) exhibits larger radius than $Zn^{2+}$ (74 pm), the hypothesis of $Mn^{4+}$ (67 pm) doping can be excluded [29,30]. Therefore, partial $Mn^{2+}$ atoms were successfully doped in the ZnO lattice, especially with a 50 at% $Mn^{2+}$ loading. Furthermore, the prepared sample was annealed and maintained in a controlled $NH_3$ atmosphere, which can avoid the $Mn^{2+}$ transfer to $Mn^{4+}$.

## 3.2. Photocatalytic degradation

The photocatalytic activity of $Mn^{2+}$-doped ZnO nanofibres under visible light ($400 < \lambda < 750$) was evaluated by degradation of RhB dye. The variation of the RhB concentration was monitored by detecting the maximum absorption peak at 554 nm using the ultraviolet–visible spectrum. The total optical power of 100 mW $ml^{-1}$ is applied to the solution, and the BET test shows an effective surface area of $17.97 \pm 2.36$ $m^2$ $g^{-1}$. Photocatalytic removal of RhB by $Mn^{2+}$-doped ZnO nanofibres under visible light is shown in figure 4a. In figure 4b, it shows a great improvement in photocatalytic activity of ZnO nanofibres when adding with $Mn^{2+}$. Fifty per cent $Mn^{2+}$ loading exhibits the best photocatalytic activity and the resulting solution became almost completely transparent after 260 min irradiating. Since theoretical calculations showed that the material exhibits the best catalytic effect when adding with 50% MnO [1], the excess MnO does not contribute much to the catalytic effect. Therefore, in this work, an adding amount of 50% is selected.

As known, the photodegradation process can be regarded as a pseudo-first-order reaction [10,29]. Based on the photocatalytic performances, its kinetics rate constant k can be calculated by the following formula:

$$C = C_0 e^{-kt},$$

where $t$ is the reaction time, $C_0$ ($2.5 \times 10^{-5}$ mol $l^{-1}$) is the initial RhB concentration, while $C$ is the RhB concentration at the time $t$, and k is the degradation rate constant. In figure 4$c$, the black slope shows the relationship between the rate constant k and the doping amount of $Mn^{2+}$. The photocatalytic efficiency of the ZnO NFs shows the best performance when adding with 50% $Mn^{2+}$. Then, we calculated the apparent quantum efficiency (AQE) using the following formula [2,3]:

$$AQE = \frac{d[x]/dt}{d[hv]_{inc}/dt} = \frac{kC_0}{TOP},$$

where $d[x]/dt$ is the initial rate of RhB concentration degradation, which can be considered as $kC_0$ in this work, while the $d[hv]_{inc}/dt$ is the total optical power (TOP) applying to the NFs. It is indicated that the photocatalytic activity of the MnO–ZnO composite shows a much better quantum efficiency up to 7.57% compared with pure ZnO (only 0.16%) under visible light. This excellent photocatalytic behaviour makes $Mn^{2+}$ loading ZnO NFs a promising visible light nanophotocatalyst. For 50% $Mn^{2+}$ loading, the catalytic activity of both degradation rate and quantum efficiency is highly improved, which can contribute to large absorption in the visible range obtained by diffuse reflectance spectra (DRS) results. Furthermore, figure 4$d$ displays the cycling performances of Mn-doped ZnO nanofibres. There is no obvious decay for the MnO–ZnO composite sample, and the photocatalytic efficiency decreases only 10% after four catalysis and drying cycles. Cyclic tests reveal that the doped ZnO nanofibres not only show good photocatalytic properties under visible light, but also can be stably re-used, which means $Mn^{2+}$-doped ZnO nanofibres are suitable for manufacturing.

## 3.3. Mechanism

A ultraviolet–visible diffuse reflection spectra at room temperature were used to investigate the bandgap energy of doped and undoped ZnO nanofibres, shown in figure 5$a$. It is worth noting that only MnO–ZnO composites exhibit large absorption in the visible range, which contribute to both doping and phase junction effects. Figure 5$b$ shows the diffuse reflection spectra and the bandgaps calculated by the Tauc plot. The bandgap of MnO–ZnO composite (2.25 eV) is much lower than pure ZnO samples, even lower than the calculated value (2.6 eV) [1]. That may due to two reasons: one is the calculated value is only related to the MnO–ZnO composite and the Mn doping effect was ignored; another is the calculated value is related to the bulk material, regardless of any size effect. Based on the XRD and TEM results, there is an observation of $Mn^{2+}$ doping and two crystalline phases of MnO and ZnO in the MnO–ZnO composite. Thus, the mechanism of photocatalytic activity in this study includes two effects, one is Mn doping effect and the other is phase junction effect [2,3]. The doping of $Mn^{2+}$ changes the band structure of ZnO, resulting in the presence of impurity levels in the band gap of ZnO. With regard to the valence band, doping of Mn causes photo-generated electrons move toward impurity levels or conductive band. Under light irradiation, the impurity level produces a large amount of superoxide radicals ($\cdot O^{2-}$), thereby promoting the level of d–d transition between impurity levels. At the same time, the photoexcited holes are transferred to the surface of the fibre after reaction with water molecules to form a highly oxidizing hydroxyl radical ($\cdot OH$).

This mechanism can be described by the following equations:

$$ZnO + hv \rightarrow e^- + h^+,$$
$$e^- + O_2 \rightarrow ZnO + \cdot O^{2-},$$
$$h^+ + H_2O \rightarrow H^+ + \cdot OH$$
$$\text{and} \quad h^+ + OH^- \rightarrow \cdot OH.$$

The optical band gap of the ZnO sample is narrowed by the doping of $Mn^{2+}$ ions and requires energy to excite electrons in the valence band. Thereby, the ZnO nanowire can smoothly perform the photocatalytic process under visible light. In addition, doped $Mn^{2+}$ ions can be used as electron traps, which can suppress the recombination process and promote the electron–hole pair separation [1,30]. Another mechanism of phase junction effects between MnO and ZnO in the composite is schematically shown in figure 6. When the photon energy is higher than the band gap of ZnO, the

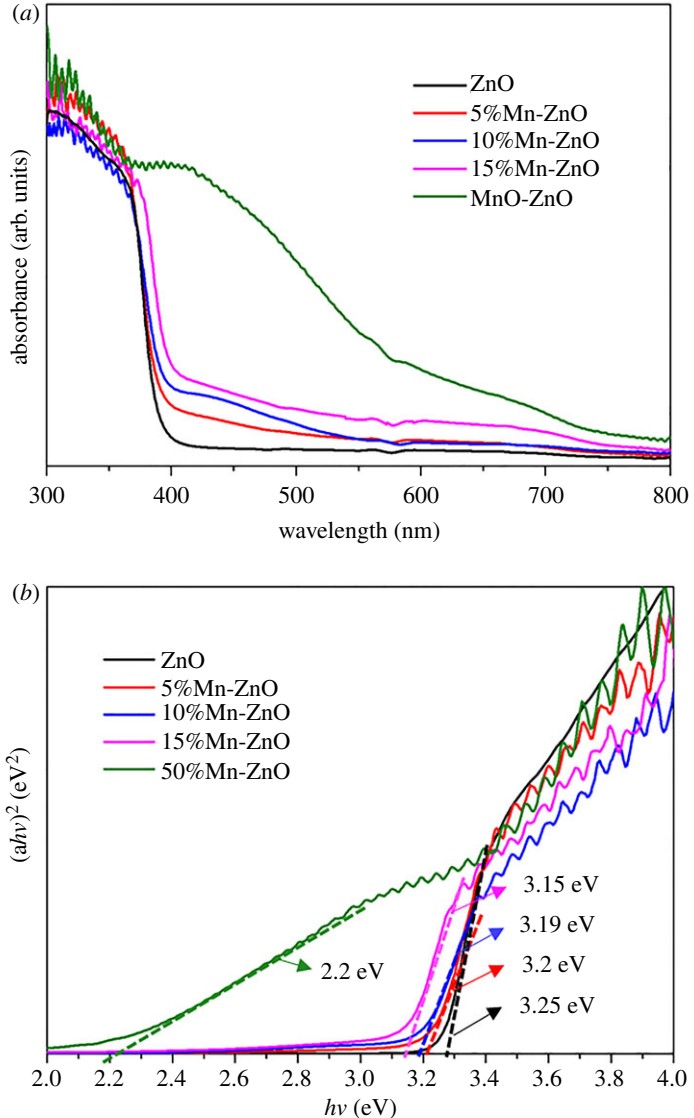

**Figure 5.** (*a*) Ultraviolet–visible diffuse reflectance spectra (DRS) of undoped and Mn²⁺-doped ZnO nanofibres. (*b*) Band gap calculation plot of $(\alpha h\nu)^2$ versus photon energy ($h\nu$).

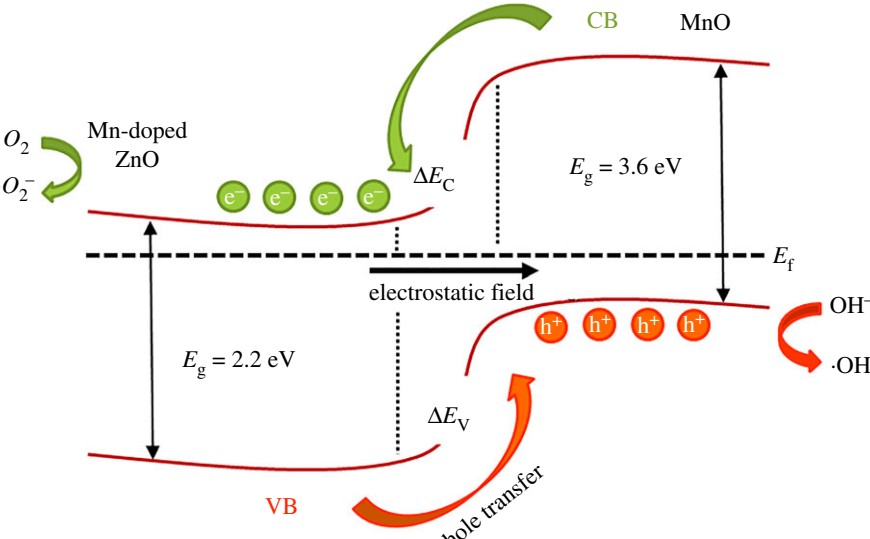

**Figure 6.** Schematic energy band diagram for MnO–ZnO nanofibre under visible light, showing charge transfer process.

electrons in the valence band are excited to the conduction band (CB) while generating the same number of holes. Due to the phase junction structure of MnO and ZnO, electrons will transfer from CB of MnO to CB of ZnO, resulting in an increase in lifetime of photo-generated electron–hole pairs [1,22]. The induced pores then react with surface $H_2O$ or $OH^-$ to form hydroxyl radicals ($\cdot OH$), which can oxidize the RhB molecules into inorganic molecules or ions [31]. The reaction process is as shown in the above formula.

# 4. Conclusion

In summary, $Mn^{2+}$-doped ZnO nanofibres were successfully prepared by electrospinning and a controlled annealing process. The MnO–ZnO composite shows much higher photocatalytic activity than pure ZnO nanofibres under visible light. Their excellent photocatalytic activity can be attributed to the $Mn^{2+}$ doping ions and the phase junction structure between ZnO and MnO. Under visible light, $Mn^{2+}$ doping helps photo-generated electrons move toward impurity levels or conductive bands, thereby reducing the band gap and promoting visible light absorption. The phase junction between MnO and $Mn^{2+}$-doped ZnO promotes the transfer of electrons from CB of MnO to CB of $Mn^{2+}$-doped ZnO, resulting in an increase in lifetime of photo-generated electron–hole pairs. These results indicate that MnO and $Mn^{2+}$ doping is a better choice for improving the photocatalytic performance of ZnO nanofibres under visible light.

Data accessibility. All the experimental data are included in the manuscript. The data are available in the Dryad Digital Repository at: https://doi.org/10.5061/dryad.nn2v75c [32].

Authors' contributions. Y.W., X.H. and Z.W. contributed to the design and the conduction of the experiment and drafting the manuscript. M.D. and L.C. contributed to the drafting of the manuscript.

Competing interests. We have no competing interests.

Funding. This research was supported by grants from the Danish National Research Foundation (grant no. DFF-6108-00396), Young Investigator Program from the Villum Foundation (grant no. VKR022954), AUFF NOVA-Project (grant no. AUFF-E-2015-FLS-9-18), EU H2020 (MNR4SCELL no. 734174), International Technological Collaboration Project of Shanghai (grant no. 17520710300), National Natural Science Foundation of China (grant no. 51671136), Research start-up funds of DGUT (grant no. GC300501-17), Fundamental Research Funds for the Central Universities, China (YJ201893) and State Key Lab of Advanced Metals and Materials, China (grant no. 2019-Z03).

Acknowledgements. All the people who contributed to the study are listed as co-authors.

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
