## [Reviewer comments · Royal Society Open Science]

Review History

RSOS-191050.R0 (Original submission)

Review form: Reviewer 1

Is the manuscript scientifically sound in its present form?

Yes

Are the interpretations and conclusions justified by the results?

Yes

Is the language acceptable?

Yes

Do you have any ethical concerns with this paper?

No

Have you any concerns about statistical analyses in this paper?

No

Recommendation?

Major revision is needed (please make suggestions in comments)

Comments to the Author(s)

I'd like to suggest authors to make some revision. Detail comments are listed below.

1.

In the page 8, the authors said "there is no MnO phase in the 0, 5, 10, 15% Mn²⁺ doping samples but a clear MnO phase peak in the Mn²⁺-doped ZnO composite", the "in the Mn²⁺-doped ZnO composite" should be revised as "in the 50% Mn²⁺-doped ZnO composite".

2.

In the Figure 4, the legend of Figure 4c is wrong and the legend of Figure 4d is missing. The legend should be revised.

3.

In the Figure 5, it will be clearer to use the same name for the green line.

4.

In the Figure 5b, the bandgap of the whole 50% Mn²⁺-doped ZnO composite is calculated as 2.25 eV. However in the Figure 6, the energy band diagram of 50% Mn²⁺-doped ZnO nanofiber is divided into Mn²⁺-doped ZnO (2.2 eV) and MnO (3.6 eV). Is it reasonable to use 2.2 eV for part of 50% Mn²⁺-doped ZnO nanofiber in energy band diagram? please explain this contradiction.

5.

In the page 11, the authors said "doped Mn²⁺ ions can be used as electron traps, which can suppress the recombination process and promote the electron-hole pair separation". The author may need offer the experimental data about charge carrier recombination of samples to prove it.

6.

In the page 12, the authors said "Mn²⁺-doped ZnO nanofibers show much higher photocatalytic activity than Mn²⁺-doped ZnO and pure ZnO nanofibers under the visible light". This sentence should be revised.

7.

Finally, I recommend some related literature to support the topic background. ACS Applied Energy Materials, DOI: 10.1021/acsaem.9b01064; APPLIED CATALYSIS B-ENVIRONMENTAL, DOI: 10.1016/j.apcatb.2017.02.049.

Review form: Reviewer 2

Is the manuscript scientifically sound in its present form?

Yes

Are the interpretations and conclusions justified by the results?

No

Is the language acceptable?

No

Do you have any ethical concerns with this paper?

No

Have you any concerns about statistical analyses in this paper?

No

Recommendation?

Major revision is needed (please make suggestions in comments)

Comments to the Author(s)

In this MS, Wang et al prepared Mn-doped ZnO photocatalyst through the electrospinning technology with subsequent annealing. The as-obtained products were characterized by XRD,

SEM, TEM and EDS mapping. It was found that the amount of Mn in ZnO could affect the photocatalytic activity of the final product. ZnO with 50% of Mn exhibited the strongest photocatalytic performance for degradation of RhB. Although some findings are interesting, the present MS cannot be accepted owing to the below reasons:

1. The authors claimed that N-doped ZnO was prepared, but no evidence proved the presence of N. In addition, the authors should experimentally prove the influence of N on the photocatalytic performance of the target catalyst.
2. The distance between Xe lamp and RhB solution should be mentioned.
3. As shown in Fig.1a, the authors claimed that as-spun Mn²⁺-doped ZnO nanofibers with a diameter of ~600 nm were prepared. However, according to experimental description the smooth nanofibers should be the precursor to obtain the final product before annealing. Here, PVP should exist in these smooth nanofibers.
4. In EDS analyses, no N element was detected. The related description in P5 is error.
5. In P6, Line 17-20: The sentence of "There is no MnO...ZnO composite." is incomplete.
6. "50% Mn-doped ZnO" was non-scientific expression. In general, the dopant is a small amount in a host. Also, ZnO and MnO were detected in the final catalyst, respectively, according to HRTEM and XRD characterization.
7. In P6, Line 27-30: The authors claimed that Mn readily donated its electrons to O, ... In this work, Mn²⁺ ions were used in fact. Were Mn ions with higher valences formed (Mn⁴⁺ ions were mentioned.)? I cannot understand the authors' meanings.
8. The authors should note the superscript and subscript.

Review form: Reviewer 3

Is the manuscript scientifically sound in its present form?

No

Are the interpretations and conclusions justified by the results?

No

Is the language acceptable?

No

Do you have any ethical concerns with this paper?

No

Have you any concerns about statistical analyses in this paper?

No

Recommendation?

Reject

Comments to the Author(s)

1. Some sentences should be rewritten, as marked in the following paragraph: In this study, we report a high efficiency photocatalyst synthesized by Mn²⁺ doped ZnO nanofibers (NFs) fabricated by facile electrospinning and a following annealing process, in which Mn²⁺ successes incorporate to ZnO NFs lattice without changing any morphology and crystalline structure of ZnO. The photodegradation properties were studied for ZnO doping with different concentrations of Mn²⁺ (5, 10, 15 and 50 at.%). The 50 % Mn²⁺-doped ZnO NFs owns excellent active photocatalytic performance (quantum efficiency up to 7.57 %) compared to pure ZnO (0.16 %) under visible light and can be considered as an efficient visible-light photocatalyst material. We systematically analyzed the catalytic mechanism and showed that the enhancement belongs to the Mn doping effect and the phase junction between MnO and ZnO. The dominant

mechanism of Mn doping leads to the presence of impurity levels in the band gap of ZnO, narrowing the optical band gap of ZnO. In addition, doped Mn²⁺ ions can be used as electron traps that inhibit the recombination process and promote electron-hole pair separation. In summary, this paper provides a convenient method for fabricating highly efficient visible light photocatalysts using controlled annealing.

Due to the low energy consumption, low cost, low toxicity and none secondary pollution, photocatalytic degradation has been considered as a promising strategy to address the current environmental issue¹⁻⁴. Photocatalysts which could effectively reduce the barrier in photocatalytic reaction is one of the key factors getting much attention^{5,6}. TiO₂ has been considered as one of photocatalyst candidate, but its low efficiency in the visible light range limit its application⁷. Although the visible light response can be achieved by doping, the efficiency is still low. Therefore, the development and design of high-efficiency new photocatalysis has become a strategy to solve the low efficiency of visible light catalysis.

There are too many sentences and they could not be listed one by one.

2. What's the meaning of PVV?

3. Is there any defined evidence to show Mn cation was doped into the lattice of ZnO crystal?

And XPS results of the samples should be provided. If 50% Mn²⁺-doped ZnO was fabricated successfully, the sample should be called as MnO₂ and ZnO composite, instead of Mn²⁺-doped ZnO.

4. What's the meaning of "The 50 % Mn²⁺-doped ZnO NFs"? Does that mean 50% of Mn in the precursor was doped into the lattice of ZnO crystal or 50% of Zn was replaced by Mn?

4. The mechanism of the paper proposed, there is also no enough evidence to support it, especially for the phase junction between MnO and Mn²⁺ doped ZnO, no microstructure of the sample to support this structure. On the other hand, if there is a junction between the two phases, MnO and Mn²⁺ doped ZnO, it is clear that no all Mn²⁺ cations were doped into the lattice of ZnO crystal. In this case, how to explain the mechanism that the authors proposed?

Decision letter (RSOS-191050.R0)

28-Oct-2019

Dear Dr Dong:

Title: Facile fabrication of Mn²⁺ doped ZnO photocatalysts by electrospinning

Manuscript ID: RSOS-191050

The editor assigned to your manuscript has now received comments from reviewers. We would like you to revise your paper in accordance with the referee and Subject Editor suggestions which can be found below (not including confidential reports to the Editor). Please note this decision does not guarantee eventual acceptance.

Please submit your revised paper before 20-Nov-2019. Please note that the revision deadline will expire at 00.00am on this date. If we do not hear from you within this time then it will be assumed that the paper has been withdrawn. In exceptional circumstances, extensions may be possible if agreed with the Editorial Office in advance. We do not allow multiple rounds of revision so we urge you to make every effort to fully address all of the comments at this stage. If deemed necessary by the Editors, your manuscript will be sent back to one or more of the original reviewers for assessment. If the original reviewers are not available we may invite new reviewers.

To revise your manuscript, log into <http://mc.manuscriptcentral.com/rsos> and enter your

Author Centre, where you will find your manuscript title listed under "Manuscripts with Decisions." Under "Actions," click on "Create a Revision." Your manuscript number has been appended to denote a revision. Revise your manuscript and upload a new version through your Author Centre.

RSC Associate Editor:
Comments to the Author:
(There are no comments.)

RSC Subject Editor:
Comments to the Author:
(There are no comments.)

Reviewers' Comments to Author:
Reviewer: 1

Comments to the Author(s)

I'd like to suggest authors to make some revision. Detail comments are listed below.

1.

In the page 8, the authors said "there is no MnO phase in the 0, 5, 10, 15% Mn²⁺ doping samples but a clear MnO phase peak in the Mn²⁺-doped ZnO composite", the "in the Mn²⁺-doped ZnO composite" should be revised as "in the 50% Mn²⁺-doped ZnO composite".

2.

In the Figure 4, the legend of Figure 4c is wrong and the legend of Figure 4d is missing. The legend should be revised.

3.

In the Figure 5, it will be clearer to use the same name for the green line.

4.

In the Figure 5b, the bandgap of the whole 50% Mn²⁺-doped ZnO composite is calculated as 2.25 eV. However in the Figure 6, the energy band diagram of 50% Mn²⁺-doped ZnO nanofiber is divided into Mn²⁺-doped ZnO (2.2 eV) and MnO (3.6 eV). Is it reasonable to use 2.2 eV for part of 50% Mn²⁺-doped ZnO nanofiber in energy band diagram? please explain this contradiction.

5.

In the page 11, the authors said "doped Mn²⁺ ions can be used as electron traps, which can suppress the recombination process and promote the electron-hole pair separation". The author may need offer the experimental data about charge carrier recombination of samples to prove it.

6.

In the page 12, the authors said "Mn²⁺-doped ZnO nanofibers show much higher photocatalytic activity than Mn²⁺-doped ZnO and pure ZnO nanofibers under the visible light". This sentence should be revised.

7.

Finally, I recommend some related literature to support the topic background. ACS Applied Energy Materials, DOI: 10.1021/acsaem.9b01064; APPLIED CATALYSIS B-ENVIRONMENTAL, DOI: 10.1016/j.apcatb.2017.02.049.

Reviewer: 2

Comments to the Author(s)

In this MS, Wang et al prepared Mn-doped ZnO photocatalyst through the electrospinning technology with subsequent annealing. The as-obtained products were characterized by XRD, SEM, TEM and EDS mapping. It was found that the amount of Mn in ZnO could affect the photocatalytic activity of the final product. ZnO with 50% of Mn exhibited the strongest photocatalytic performance for degradation of RhB. Although some findings are interesting, the present MS cannot be accepted owing to the below reasons:

1. The authors claimed that N-doped ZnO was prepared, but no evidence proved the presence of N. In addition, the authors should experimentally prove the influence of N on the photocatalytic performance of the target catalyst.
2. The distance between Xe lamp and RhB solution should be mentioned.
3. As shown in Fig.1a, the authors claimed that as-spun Mn²⁺-doped ZnO nanofibers with a diameter of ~600 nm were prepared. However, according to experimental description the smooth nanofibers should be the precursor to obtain the final product before annealing. Here, PVP should exist in these smooth nanofibers.
4. In EDS analyses, no N element was detected. The related description in P5 is error.
5. In P6, Line 17-20: The sentence of "There is no MnO...ZnO composite." is incomplete.
6. "50% Mn-doped ZnO" was non-scientific expression. In general, the dopant is a small amount in a host. Also, ZnO and MnO were detected in the final catalyst, respectively, according to HRTEM and XRD characterization.
7. In P6, Line 27-30: The authors claimed that Mn readily donated its electrons to O, ... In this work, Mn²⁺ ions were used in fact. Were Mn ions with higher valences formed (Mn⁴⁺ ions were mentioned.)? I cannot understand the authors' meanings.
8. The authors should note the superscript and subscript.

Reviewer: 3

Comments to the Author(s)

1. Some sentences should be rewritten, as marked in the following paragraph:

In this study, we report a high efficiency photocatalyst synthesized by Mn²⁺ doped ZnO nanofibers (NFs) fabricated by facile electrospinning and a following annealing process, in which Mn²⁺ successes incorporate to ZnO NFs lattice without changing any morphology and crystalline structure of ZnO. The photodegradation properties were studied for ZnO doping with different concentrations of Mn²⁺ (5, 10, 15 and 50 at.%). The 50 % Mn²⁺-doped ZnO NFs owns

excellent active photocatalytic performance (quantum efficiency up to 7.57 %) compared to pure ZnO (0.16 %) under visible light and can be considered as an efficient visible-light photocatalyst material. We systematically analyzed the catalytic mechanism and showed that the enhancement belongs to the Mn doping effect and the phase junction between MnO and ZnO. The dominant mechanism of Mn doping leads to the presence of impurity levels in the band gap of ZnO, narrowing the optical band gap of ZnO. In addition, doped Mn²⁺ ions can be used as electron traps that inhibit the recombination process and promote electron-hole pair separation. In summary, this paper provides a convenient method for fabricating highly efficient visible light photocatalysts using controlled annealing.

Due to the low energy consumption, low cost, low toxicity and none secondary pollution, photocatalytic degradation has been considered as a promising strategy to address the current environmental issue¹⁻⁴. Photocatalysts which could effectively reduce the barrier in photocatalytic reaction is one of the key factors getting much attention^{5,6}. TiO₂ has been considered as one of photocatalyst candidate, but its low efficiency in the visible light range limit its application⁷. Although the visible light response can be achieved by doping, the efficiency is still low. Therefore, the development and design of high-efficiency new photocatalysis has become a strategy to solve the low efficiency of visible light catalysis.

There are too many sentences and they could not be listed one by one.

2. What's the meaning of PVV?

3. Is there any defined evidence to show Mn cation was doped into the lattice of ZnO crystal? And XPS results of the samples should be provided. If 50% Mn²⁺-doped ZnO was fabricated successfully, the sample should be called as MnO₂ and ZnO composite, instead of Mn²⁺-doped ZnO.

4. What's the meaning of "The 50 % Mn²⁺-doped ZnO NFs"? Does that mean 50% of Mn in the precursor was doped into the lattice of ZnO crystal or 50% of Zn was replaced by Mn?

4. The mechanism of the paper proposed, there is also no enough evidence to support it, especially for the phase junction between MnO and Mn²⁺ doped ZnO, no microstructure of the sample to support this structure. On the other hand, if there is a junction between the two phases, MnO and Mn²⁺ doped ZnO, it is clear that no all Mn²⁺ cations were doped into the lattice of ZnO crystal. In this case, how to explain the mechanism that the authors proposed?

Author's Response to Decision Letter for (RSOS-191050.R0)

See Appendix A.

RSOS-191050.R1 (Revision)

Review form: Reviewer 3

Is the manuscript scientifically sound in its present form?

Yes

Are the interpretations and conclusions justified by the results?

Yes

Is the language acceptable?

Yes

Do you have any ethical concerns with this paper?

No

Have you any concerns about statistical analyses in this paper?

No

Recommendation?

Accept as is

Comments to the Author(s)

No comments

Decision letter (RSOS-191050.R1)

06-Jan-2020

Dear Dr Dong:

Title: Facile fabrication of Mn²⁺ doped ZnO photocatalysts by electrospinning
Manuscript ID: RSOS-191050.R1

It is a pleasure to accept your manuscript in its current form for publication in Royal Society Open Science. The chemistry content of Royal Society Open Science is published in collaboration with the Royal Society of Chemistry.

RSC Associate Editor:
Comments to the Author:
I apologise that this has taken longer than usual.

RSC Subject Editor:
Comments to the Author:
(There are no comments.)

Reviewer(s)' Comments to Author:
Reviewer: 3

Comments to the Author(s)
No comments

Appendix A

Reviewers' Comments to Author

We would like to thank the editor's effort and referees' very constructive suggestions. Here, we have revised the manuscript according to the comments by point-by-point.

Reviewer: 1

I'd like to suggest authors to make some revision. Detail comments are listed below.

Comment 1: In the page 8, the authors said "there is no MnO phase in the 0, 5, 10, 15% Mn²⁺ doping samples but a clear MnO phase peak in the Mn²⁺-doped ZnO composite", the "in the Mn²⁺-doped ZnO composite" should be revised as "in the 50% Mn²⁺-doped ZnO composite".

Response 1:

We thank for referee's very carefully reviewing. We are so sorry for our carelessness. In this revision, we have revised the text to "There is no MnO phase in the 0, 5, 10, 15% Mn²⁺ doping samples, but there was a clear MnO phase peak in the MnO-ZnO composite." on Page 6, Line 7.

Comment 2 In the Figure 4, the legend of Figure 4c is wrong and the legend of Figure 4d is missing. The legend should be revised.

Response 2:

Thank for referee's kind comment. In this revision, we have revised the legend of Figure 4c and added the legend of Figure 4d.

Comment 3 In the Figure 5, it will be clearer to use the same name for the green line.

Response 3:

Many thanks for referee's good suggestion. In this revision, we have modified the title of the green line in Figure 5.

Comment 4 In the Figure 5b, the bandgap of the whole 50% Mn²⁺-doped ZnO composite is calculated as 2.25 eV. However in the Figure 6, the energy band diagram of 50% Mn²⁺-doped ZnO nanofiber is divided into Mn²⁺-doped ZnO (2.2 eV) and MnO (3.6 eV). Is it reasonable to use 2.2 eV for part of 50% Mn²⁺-doped ZnO nanofiber in energy band diagram? please explain this contradiction.

Response 4:

Thank you for the comment. The whole 50% Mn²⁺-doped ZnO composite is calculated as 2.25 eV, thus, the part of 50% Mn²⁺-doped ZnO nanofiber is not 2.25 eV.

We are sorry for the inaccuracy and we revised the mechanism of the electron transfer part to ZnO and MnO junction. We have revised the band gap of ZnO nanofiber to 3.25 eV in Figure 6.

Comment 5 In the page 11, the authors said “doped Mn²⁺ ions can be used as electron traps, which can suppress the recombination process and promote the electron-hole pair separation”. The author may need offer the experimental data about charge carrier recombination of samples to prove it.

Response 5:

We thank for referee’s good suggestion. We fully agree with referee that additional experiment data about charge carrier recombination will make sense. However, currently we are unable to perform this experiment because it is still a challenge to accurately display the charge carrier recombination on power-like sample. Instead of it, in this revision, we have provide a few theoretical literatures (Ref. 1 and 32) where the suppressing the recombination by the doped ions has been demonstrated. The text has been revised on Page 9, Last 4 Line.

Comment 6 In the page 12, the authors said “Mn²⁺-doped ZnO nanofibers show much higher photocatalytic activity than Mn²⁺-doped ZnO and pure ZnO nanofibers under the visible light”. This sentence should be revised.

Response 6:

Thanks referee for the good suggestion. In this revision, we have corrected the sentence to “The MnO-ZnO composite show much higher photocatalytic activity than pure ZnO nanofibers under the visible light.” on Page 9 Line 6.

Comment 7. Finally, I recommend some related literature to support the topic background. ACS Applied Energy Materials, DOI: 10.1021/acsaem.9b01064;; APPLIED CATALYSIS B-ENVIRONMENTAL, DOI: 10.1016/j.apcatb.2017.02.049.

Response 7:

We thank referee for the excellent literatures. The suggested literatures demonstrated a facile ion-replacement strategy for fabricating ZnO/ZnSe/CdSe/Cu_{2-x}Se core-shell nanowire arrays and highly efficient electrodes for water splitting, which is well inline with this manuscript. In this revision, the suggested literatures have been cited as Ref. 8 and 9, and the main text has been revised on Page 2 Line 5 and Line8.

Reviewer: 2

In this MS, Wang et al prepared Mn-doped ZnO photocatalyst through the

electrospinning technology with subsequent annealing. The as-obtained products were characterized by XRD, SEM, TEM and EDS mapping. It was found that the amount of Mn in ZnO could affect the photocatalytic activity of the final product. ZnO with 50% of Mn exhibited the strongest photocatalytic performance for degradation of RhB. Although some findings are interesting, the present MS cannot be accepted owing to the below reasons:

Comment 1 The authors claimed that N-doped ZnO was prepared, but no evidence proved the presence of N. In addition, the authors should experimentally prove the influence of N on the photocatalytic performance of the target catalyst.

Response 1:

Thanks for referee's good comments. We have carefully checked the data, and we apologize that there is not nitrogen atoms doped in ZnO. We are so sorry for our carelessness. In this revision, the main text has been carefully revised accordingly.

Comment 2 The distance between Xe lamp and RhB solution should be mentioned.

Response 2:

Thanks for the good suggestion. In this revision, the detailed parameters have been provided in the experiment section in Paragraph 1, Page 5.

Comment 3 As shown in Fig.1a, the authors claimed that as-spun Mn²⁺-doped ZnO nanofibers with a diameter of ~600 nm were prepared. However, according to experimental description the smooth nanofibers should be the precursor to obtain the final product before annealing. Here, PVP should exist in these smooth nanofibers.

Response 3:

Thank you for the good comment. Indeed, the SEM image in Figure 1 shows the as-spun nanofibers before annealing. Yes, the PVP is present in these nanofibers. We are so sorry for our carelessness. In this revision, we have revised the sentence to "Figure 1a shows a SEM image of as-spun nanofibers with a diameter about 600 nm." on Page 5 Line 8.

Comment 4. In EDS analyses, no N element was detected. The related description in P5 is error.

Response 4:

We are so sorry for our carelessness. In this revision, we have corrected the description part to "Energy dispersive X-ray spectroscopy (EDS) analysis confirmed that ZnO, O, Mn elements were uniformly present in the nanofibers, and the ratio of M and O elements was approximately 1:1 (Fig. 2b)." in Page 5.

Comment 5. In P6, Line 17-20: The sentence of "There is no MnO...ZnO composite." is incomplete.

Response 5:

Thanks for referee's carefully reviewing. We have modified the sentence to "There is no MnO phase in the 0, 5, 10, 15% Mn²⁺ doping samples, but there was a clear MnO phase peak in the 50% Mn²⁺-doped ZnO composite." on Page 6 Line 7.

Comment 6. "50% Mn-doped ZnO" was non-scientific expression. In general, the dopant is a small amount in a host. Also, ZnO and MnO were detected in the final catalyst, respectively, according to HRTEM and XRD characterization.

Response 6:

Thank you for the comment. We have re-defined the expression of our samples and revised "Mn²⁺ doped ZnO NFs" to "ZnO NFs added Mn²⁺" in Page 4, Line 3, and "50% Mn-doped ZnO" to "MnO-ZnO composite" in the manuscript. All the revised parts are marked by red.

Comment 7. In P6, Line 27-30: The authors claimed that Mn readily donated its electrons to O, ... In this work, Mn²⁺ ions were used in fact. Were Mn ions with higher valences formed (Mn⁴⁺ ions were mentioned.)? I cannot understand the authors' meanings.

Response 7:

Thanks for referee's good comments. We are so sorry for our carelessness which makes the reader confuse. Actually, there is no Mn⁴⁺ ion formed based on our measurements. Moreover, the sample has been annealed under a reduced gas to avoid the Mn²⁺ transfer to Mn⁴⁺. To clarify it, we have added more explanation in Page 6, Paragraph 1. "Furthermore, the prepared sample was annealed and maintained in a controlled NH₃ atmosphere, which can avoid the Mn²⁺ transfer to Mn⁴⁺."

Comment 8. The authors should note the superscript and subscript.

Response 8:

We thank referee's good suggestions. We have carefully checked and revised the superscript and subscript in the manuscript.

Reviewer: 3

Comment 1. Some sentences should be rewritten, as marked in the following paragraph:

In this study, we report a high efficiency photocatalyst synthesized by Mn²⁺ doped

ZnO nanofibers (NFs) fabricated by facile electrospinning and a following annealing process, in which Mn²⁺ successively incorporate to ZnO NFs lattice without changing any morphology and crystalline structure of ZnO. The photodegradation properties were studied for ZnO doping with different concentrations of Mn²⁺ (5, 10, 15 and 50 at.%). The 50 % Mn²⁺-doped ZnO NFs owns excellent active photocatalytic performance (quantum efficiency up to 7.57 %) compared to pure ZnO (0.16 %) under visible light and can be considered as an efficient visible-light photocatalyst material. We systematically analyzed the catalytic mechanism and showed that the enhancement belongs to the Mn doping effect and the phase junction between MnO and ZnO. The dominant mechanism of Mn doping leads to the presence of impurity levels in the band gap of ZnO, narrowing the optical band gap of ZnO. In addition, doped Mn²⁺ ions can be used as electron traps that inhibit the recombination process and promote electron-hole pair separation. In summary, this paper provides a convenient method for fabricating highly efficient visible light photocatalysts using controlled annealing.

Due to the low energy consumption, low cost, low toxicity and none secondary pollution, photocatalytic degradation has been considered as a promising strategy to address the current environmental issue^{1–4}. Photocatalysts which could effectively reduce the barrier in photocatalytic reaction is one of the key factors getting much attention^{5,6}. TiO₂ has been considered as one of photocatalyst candidate, but its low efficiency in the visible light range limit its application⁷. Although the visible light response can be achieved by doping, the efficiency is still low. Therefore, the development and design of high-efficiency new photocatalysis has become a strategy to solve the low efficiency of visible light catalysis.

There are too many sentences and they could not be listed one by one.

Response 1:

We have carefully revised the marked part and whole manuscript. All the revised parts are marked by red.

" In this study, a high efficiency photocatalyst was synthesized by Mn²⁺ doped ZnO nanofibers (NFs) fabricated by facile electrospinning and a following annealing process, in which Mn²⁺ successively incorporate to ZnO NFs lattice without changing any morphology and crystalline structure of ZnO. The photodegradation properties of ZnO doping with different concentrations of Mn²⁺ (5, 10, 15 and 50 at.%) were investigated. The 50 % MnO-ZnO composite owns excellent active photocatalytic performance (quantum efficiency up to 7.57 %) compared to pure ZnO (0.16 %) under visible light and can be considered as an efficient visible-light photocatalyst material. We systematically analyzed its catalytic mechanism and found that the enhancement belongs to the Mn doping effect and the phase junction between MnO and ZnO. The dominant mechanism of Mn doping leads to the presence of impurity

levels in the band gap of ZnO, narrowing the optical band gap of ZnO. In addition, doped Mn²⁺ ions can be used as electron traps that inhibit the recombination process and promote electron-hole pair separation. In summary, this paper provides a convenient method for fabricating highly efficient visible light photocatalysts using controlled annealing."

"Due to the low energy consumption, low cost, low toxicity and none secondary pollution, photocatalytic degradation has been considered as a promising strategy to address the current environmental issue¹⁻⁴. Photocatalysts that can effectively reduce the barrier in photocatalytic reactions have attracted attention^{5,6}. TiO₂ has been considered as one of photocatalyst candidate, but its low efficiency in the visible light range limits its application⁷. Although the visible light response of TiO₂-based catalysts can be improved by doping, the efficiency is still low. Therefore, the development and design of high-efficiency new photocatalysis has become a strategy to solve the low efficiency of visible light catalysis⁸. "

Comment 2. What's the meaning of PVP?

Response 2:

We are so sorry for our carelessness. The PVP is polyvinyl pyrrolidone. Now, we have given the full name in its first appearance on Page 4, Line 5. Besides, we have also carefully checked the whole manuscript.

Comment 3. Is there any defined evidence to show Mn cation was doped into the lattice of ZnO crystal? And XPS results of the samples should be provided. If 50% Mn²⁺-doped ZnO was fabricated successfully, the sample should be called as MnO₂ and ZnO composite, instead of Mn²⁺-doped ZnO.

Response 3:

We thank for referee's good comments. Based on the XRD and TEM results, there is an observation of Mn²⁺ doping and two crystallize phases of MnO and ZnO in the MnO-ZnO composite. We have discussed it in Page 8, Paragraph 2. Based on referee's comments, we have renamed "The 50 % Mn²⁺-doped ZnO NFs" to "MnO-ZnO composite" to avoid misunderstanding, and all the revised parts are marked by red.

Comment 4. What's the meaning of "The 50 % Mn²⁺-doped ZnO NFs"? Does that mean 50% of Mn in the precursor was doped into the lattice of ZnO crystal or 50% of Zn was replaced by Mn?

Response 4:

We thank for referee's good comments. "The 50 % Mn²⁺-doped ZnO NFs" means 50%

of Mn in the precursor, but part of the Mn doped into the lattice of ZnO crystal. Based on your comments, we have revised “The 50 % Mn²⁺-doped ZnO NFs” to “MnO-ZnO composite” to avoid misunderstanding.

Comment 5. The mechanism of the paper proposed, there is also no enough evidence to support it, especially for the phase junction between MnO and Mn²⁺ doped ZnO, no microstructure of the sample to support this structure. On the other hand, if there is a junction between the two phases, MnO and Mn²⁺ doped ZnO, it is clear that not all Mn²⁺ cations were doped into the lattice of ZnO crystal. In this case, how to explain the mechanism that the authors proposed?

Response 5:

Thank you for your comment. We added the phase junction between MnO and ZnO in the inset of Figure 2c, and more discussions in Page 6, Paragraph 1. Based on your second comment, we have modified the Figure 6 and the relevant discussion sections in Page 9, last paragraph. We have changed the Mn²⁺ doped ZnO to ZnO and simplified the electron transfer mechanism of the phase junction between MnO and ZnO. We hope this revision can meet with approval.